# Gamma-Glutamyl Transferase Plus Carcinoembryonic Antigen Ratio Index: A Promising Biomarker Associated with Treatment Response to Neoadjuvant Chemotherapy for Patients with Colorectal Cancer Liver Metastases

**DOI:** 10.3390/curroncol32020117

**Published:** 2025-02-18

**Authors:** Yanjiang Yin, Bowen Xu, Jianping Chang, Zhiyu Li, Xinyu Bi, Zhicheng Wei, Xu Che, Jianqiang Cai

**Affiliations:** 1Department of Hepatobiliary Surgery, National Cancer Center/National Clinical Research Center for Cancer/Cancer Hospital, Chinese Academy of Medical Sciences and Peking Union Medical College, Beijing 100021, China; bruceyyj@foxmail.com (Y.Y.); xubowen1112@126.com (B.X.); chang_jianp123@126.com (J.C.); lizhiyu008@126.com (Z.L.); beexy71@126.com (X.B.); 2Key Laboratory of Gene Editing Screening and Research and Development (R&D) of Digestive System Tumor Drugs, National Cancer Center/National Clinical Research Center for Cancer/Cancer Hospital, Chinese Academy of Medical Sciences and Peking Union Medical College, Beijing 100021, China; 3Department of Hepatobiliary Surgery, General Surgery, Qilu Hospital, Cheeloo College of Medicine, Shandong University, Jinan 250012, China; 4Department of Hepatobiliary and Pancreatic Surgery, National Cancer Center/National Clinical Research Center for Cancer/Cancer Hospital & Shenzhen Hospital, Chinese Academy of Medical Sciences and Peking Union Medical College, Shenzhen 518116, China; drweizhicheng@126.com

**Keywords:** colorectal cancer liver metastases, neoadjuvant chemotherapy, biomarker, machine learning, treatment response

## Abstract

Background: Colorectal cancer liver metastasis (CRLM) is a significant contributor to cancer-related illness and death. Neoadjuvant chemotherapy (NAC) is an essential treatment approach; however, optimal patient selection remains a challenge. This study aimed to develop a machine learning-based predictive model using hematological biomarkers to assess the efficacy of NAC in patients with CRLM. Methods: We retrospectively analyzed the clinical data of 214 CRLM patients treated with the XELOX regimen. Blood characteristics before and after NAC, as well as the ratios of these biomarkers, were integrated into the machine learning models. Logistic regression, decision trees (DTs), random forest (RF), support vector machine (SVM), and AdaBoost were used for predictive modeling. The performance of the models was evaluated using the AUROC, F1-score, and external validation. Results: The DT (AUROC: 0.915, F1-score: 0.621) and RF (AUROC: 0.999, F1-score: 0.857) models demonstrated the best predictive performance in the training cohort. The model incorporating the ratio of post-treatment to pre-treatment gamma-glutamyl transferase (rGGT) and carcinoembryonic antigen (rCEA) formed the GCR index, which achieved an AUROC of 0.853 in the external validation. The GCR index showed strong clinical relevance, predicting better chemotherapy responses in patients with lower rCEA and higher rGGT levels. Conclusions: The GCR index serves as a predictive biomarker for the efficacy of NAC in CRLM, providing a valuable clinical reference for the prognostic assessment of these patients.

## 1. Introduction

Colorectal cancer ranks as the third most frequently diagnosed cancer globally and is the second leading cause of cancer-related mortality [1]. Because the portal venous system collects mesenteric blood vessels that drain into the liver, approximately 50% of patients with colorectal cancer develop liver metastases within the first three years following their initial diagnosis [2]. Of these patients, 20% present with synchronous colorectal cancer liver metastases (CRLMs) at the time of initial diagnosis, and this subgroup of patients generally has a poor prognosis [3,4]. Although surgical treatment remains the foundation for managing CRLM, it is only applicable when the metastases are relatively isolated or confined to a single liver lobe or segment. Additionally, a significant proportion of patients experience recurrence within a year after surgery, underscoring the limitations of relying solely on surgical intervention for the management of CRLM [5]. Consequently, the relevance of perioperative chemotherapy is becoming increasingly obvious.

Neoadjuvant chemotherapy (NAC) is a treatment strategy that utilizes preoperative chemotherapy to reduce tumor burden, decrease tumor size, and increase the probability of achieving R0 resection. This approach can lower the probability of tumor recurrence after radical surgery and extend the recurrence-free survival (RFS) and overall survival (OS) [6]. Currently, there is no consensus regarding the optimal number of NAC cycles. Although NAC may achieve tumor downstaging and pathological complete response (pCR), excessive NAC cycles may increase the risk of chemotherapy-induced liver damage, postoperative complications, drug resistance, and mortality [7].

Thus, NAC functions as a double-edged sword, making it particularly important in clinical practice to identify patients who are suitable for NAC and those who should proceed directly with surgery. For patients showing a good response, it is important to avoid excessive neoadjuvant treatment, which could lead to liver damage. Conversely, for initially resectable patients with poor response to NAC, it is essential to proceed with surgery promptly to prevent tumor progression, which could eliminate the chance for surgical intervention. In clinical practice, the efficacy of NAC is typically evaluated using imaging-based RECIST criteria. But these criteria show significant heterogeneity compared to postoperative pathological responses.

The biological characteristics of tumors are strongly associated with tumor activity and the efficacy of anticancer treatments. Hematological features are the most accessible biomarkers that can reflect the biological properties of tumors and the tumor microenvironment. Machine learning has demonstrated significant application value in clinical diagnosis and the prediction of patient prognosis. In the field of colorectal cancer, machine learning has long been applied to disease etiology analysis [8,9], pathological slide diagnosis [10], and survival analysis [11,12]. Machine learning can fully utilize large datasets for training and data mining, thereby enhancing data utilization and making it particularly suitable for analyzing hematological characteristics.

In this study, we employed a supervised machine learning approach for the first time to fit the hematological biomarkers of patients before or after NAC, along with the ratio of post-treatment to pre-treatment hematological biomarkers and epidemiological characteristics, to construct a predictive model. We used the postoperative pathological treatment response as a reference to evaluate the predictive value of this model for the efficacy of NAC in patients with CRLM.

## 2. Materials and Methods

### 2.1. Study Design

We collected clinical data from 349 patients with CRLM who underwent NAC with the XELOX regimen and subsequent liver metastasis resection at the Cancer Hospital of the Chinese Academy of Medical Sciences (Center 1, n = 284) and Qilu Hospital of Shandong University (Center 2, n = 65) between January 2010 and November 2020. The inclusion criteria for patients were as follows: (a) the primary tumor was colorectal cancer, with a pathological diagnosis of liver lesions as CRLM; (b) both the primary tumor and liver metastases were successfully resected; (c) the pathological results reported the chemotherapy response of liver metastases (tumor regression grade, TRG); and (d) complete clinical hematological data were available. Finally, 135 patients who did not meet the inclusion criteria were excluded. This study was reviewed and approved by the Institutional Review Boards of both medical centers and was conducted in accordance with the Declaration of Helsinki [13]. The flowchart of the study is shown in Figure 1.

### 2.2. Diagnostic Feature Selection

We selected 21 blood features (BFs) based on experience and previous studies, including complete blood counts, biochemical indicators, electrolytes, and coagulation function parameters. We then statistically analyzed the BFs of the patients before neoadjuvant treatment with the XELOX regimen (bBFs) and after neoadjuvant treatment (aBFs), as well as the ratio of post-treatment to pre-treatment BFs (rBFs). The selected BFs included hemoglobin (Hb), platelets (PLTs), albumin (ALB), white blood cell (WBC) count, absolute neutrophil count (ANC), absolute lymphocyte count (ALC), absolute monocyte count (AMC), plasma D-dimer, lactate dehydrogenase (LDH), alanine aminotransferase (ALT), aspartate aminotransferase (AST), total bilirubin (TBIL), direct (conjugated) bilirubin (DBIL), gamma-glutamyl transferase (GGT), alkaline phosphatase (ALP), carbohydrate antigen 19-9 (CA 19-9), carcinoembryonic antigen (CEA), prognostic nutritional index (PNI), neutrophil-to-lymphocyte ratio (NLR), lymphocyte-to-monocyte ratio (LMR), and platelet-to-lymphocyte ratio (PLR). To develop a blood risk score, we incorporated these BFs along with body mass index (BMI), age, sex, number of liver metastases, preoperative comorbidities, and primary tumor location into our model for training.

### 2.3. Supervised Machine Learning

To balance simplicity and accuracy, we selected five commonly used machine learning models for analysis: logistic regression, decision tree, random forest (RF), support vector machine (SVM), and adaptive boost (AdaBoost). Data from Center 1 were randomly divided into two sets—a training set (n = 119) and a validation set (n = 30)—for model development. And data from Center 2 were used as an external validation cohort (n = 65). The success of the random allocation was confirmed by comparing the baseline characteristics of the training, validation, and test sets, ensuring no significant differences in patient characteristics across the groups. Hyperparameter tuning was conducted using a grid search to optimize the performance of each classification model. For example, in the RF model, we adjusted the number of decision trees, the maximum depth, and the minimum number of samples required for splitting. For the SVM model, we fine-tuned the regularization parameter (C) and choice of the kernel function. All models were evaluated using 10-fold cross-validation of the training and validation sets to prevent overfitting and ensure model generalizability.

### 2.4. Evaluation of Machine Learning Models

The performance of the models was assessed using several evaluation metrics, including accuracy, area under the receiver operating characteristic curve (AUROC), precision, recall, F1-score, and area under the precision–recall curve (AUPR). The final performance of each model was independently evaluated on the test set, and the best model was determined by comparing the AUROC, AUPR, and F1-score. To understand the model’s predictions, we utilized Shapley Additive Explanations (SHAPs), a technique that assesses the contribution of each feature to the model’s output. We determined the relative significance of each characteristic in the prediction of chemotherapy outcomes by calculating SHAP values. By calculating the SHAP values for each sample in the test set, we identified the clinical and biological features that were most influential in predicting chemotherapy response.

### 2.5. Statistical Analysis

The median and interquartile range (IQR) were used to describe non-normally distributed continuous data. Categorical variables were calculated using both frequencies and percentages. Differences in continuous variables between the training, validation, and test sets were compared using the Mann–Whitney U-test, and differences in categorical variables were compared by the Chi-square test or Fisher’s exact test. All statistical analyses were performed using RStudio version 1.4.1717 (Integrated Development Environment for R, Boston, MA, USA) and SPSS version 26.0 (IBM Corp., Armonk, NY, USA). All tests were two-tailed, and *p* values < 0.05 were considered statistically significant.

## 3. Results

### 3.1. Baseline Characteristics

Based on the inclusion criteria, we enrolled 149 CRLM patients from the Cancer Hospital of the Chinese Academy of Medical Sciences (Center 1, C1) and 65 CRLM patients from Qilu Hospital of Shandong University (Center 2, C2). All 214 patients received an XELOX NAC regimen. The average ages of patients in the two centers were 58.8 ± 9.37 years (C1) and 55 ± 9.28 years (C2). A comparison of the baseline characteristics between the two centers is illustrated in Table 1, showing no statistically significant differences (*p* > 0.05). To ensure comparability between the different TRG assessment tools used in the two centers (Dowrak TRG for C1 and Mandard TRG for C2), we modified the TRG assessment tool (see Appendix A). We redefined the scores as 1–5 from no response to complete remission and considered “severe treatment response” and “complete remission” (scores 4 and 5) as effective treatment, with the remaining cases (scores 1–3) defined as ineffective. According to the unified modified TRG standards, 28 patients (18.8%) had good chemotherapy responses in C1 and 14 (21.8%) had good responses in C2, with no significant difference (*p* = 0.604). We also analyzed the relationship between the chemotherapy response, patient survival, and recurrence in Center 1. The Kaplan–Meier curves indicated that the response group had significantly better OS (HR = 6.98, 95% CI = 3.574–13.632, LogRank *p* = 0.002) and RFS (HR = 2.32, 95% CI = 1.534–3.509, log-rank *p* = 0.001) than the non-response group (Figure 2A,B). Before applying machine learning, we divided the C1 data into a training set (n = 119) and validation set (n = 30), with no statistically significant differences in baseline characteristics or laboratory results between the two groups (*p* > 0.05) (Appendix A).

### 3.2. Variable Selection and Machine Learning Model Construction

In the training and validation sets, we performed 10-fold cross-validation using LASSO regression analysis on 63 blood features and six clinical characteristics available before chemotherapy. The minimum mean square error was 0.94 (Figure 2C,D), resulting in the selection of 15 important feature variables (Appendix A) for subsequent machine learning models (AdaBoost), which incorporated variable selection functionality from a total of 69 feature variables.

We trained the models using RF, SVM, logistic regression, decision trees, and AdaBoost, incorporating 10-fold cross-validation to minimize the risk of overfitting and ensure model robustness. The logistic regression model identified four independent factors predictive of NAC, namely GGT, aLMR, aCEA, and the number of metastatic tumors (Appendix A). The model’s performance was evaluated using the AUROC (Figure 3A,B). The AUC for the prediction set was 0.77 (95% CI: 0.67–0.86, DeLong), while the AUC for the validation set was 0.818 (95% CI: 0.59–1, DeLong), demonstrating the good stability of the model (Figure 3C,D). The results for the testing efficiency of the other four machine learning models are shown in Table 2. In the training set, the decision tree model made decisions with three leaf nodes (Appendix A) and ranked second, with an F1-score of 0.621 (Figure 3E) and an AUROC of 0.915 (Figure 3I). The RF model ranked first with an F1-score of 0.857 (Figure 3F) and an AUROC of 0.999 (Figure 3J). Although the SVM model had an AUROC of 0.930 (second highest, Appendix A), its F1-score was the lowest at 0.4 (Figure 3G). The AdaBoost model had the lowest AUROC (0.774) (Appendix A) and an F1-score (0.506) ranking third (Figure 3H). In the validation set, the DT and RF models ranked first and second, respectively, while the SVM and AdaBoost models performed poorly. Given the potential for overfitting in the RF model, we aimed to develop a practical and interpretable model based on the DT model by incorporating the key features that were highly important in the RF model.

### 3.3. Optimal Model Construction

Using the SHAP analysis on the training set, we conducted an interpretability analysis of the RF model to visually represent the impact of all variables. The colors represent variable values: red pixels indicate positive SHAP values that increase class likelihood, whereas blue pixels indicate negative SHAP values that decrease class probability. The bar chart illustrates the relationship between feature values and their predictive impact (Figure 4A). The results showed that the ratio of the GGT value after NAC to that before NAC (rGGT, X57) and the ratio of the CEA value after NAC to that before NAC (rCEA, X63) had the highest average absolute impact on the model output (Figure 4B).

We combined the DT and RF models to establish a GCR model, represented as GCR = f(x) + g(y) = H (x57 − 1.2) + H (0.43 − y), where x = rGGT and y = rCEA. Here, H(z) = 1 when z ≥ 0; otherwise, H(z) = 0 (Figure 4C). We performed 10-fold cross-validation of the GCR score model analysis. The average AUROC for the training set was 0.843 ± 0.032 (Figure 4D,E), while the average AUROC of the internal validation set was 0.809 ± 0.032 (Figure 4F,G). The AUROC for the validation set was lower than that for the test set, indicating good model generalizability and successful fitting [14]. External validation yielded an AUROC of 0.853 (95% CI = 0.734–0.971) (Figure 4H,I). This model demonstrates excellent diagnostic efficacy and has high interpretability, indicating its significant clinical application value.

## 4. Discussion

An increasing number of studies suggest that tumor biological characteristics are promising prognostic factors compared to tumor burden. This is because tumor biology better reflects tumor activity, metabolic capacity, and malignancy [15,16]. Currently, the RECIST criteria are the only widely accepted method for assessing tumor response to NAC [17]. However, RECIST reflects the general characteristics of solid tumor responses to chemotherapy without distinguishing between tumor types, and discrepancies often exist between radiological assessments of solid tumors and pathological findings after surgery. For instance, many tumors show reduced size on imaging but remain highly active overall. Lau et al. found that metabolic response was a more effective predictor of prognosis in patients with CRLM undergoing NAC compared to radiological response [16]. Moreover, the increasing use of biological agents, such as bevacizumab and immunotherapy, in the neoadjuvant treatment of CRLM can lead to increased tumor necrosis without significant changes in tumor size, thereby affecting the assessment using RECIST criteria [18,19].

Hematological parameters have the advantage of being readily accessible and can dynamically reflect changes in patients’ conditions during NAC treatment. If chemotherapy response could be predicted based on certain hematological parameters, it would enable more timely and personalized treatment guidance for patients. Numerous studies have attempted to predict NAC efficacy in cancer using easily accessible hematological parameters, but such studies are scarce in CRLM [18]. Furthermore, many studies focus on baseline hematological parameters before NAC or preoperative parameters after NAC, neglecting the dynamic changes in these characteristics during treatment [20]. Our study is the first to incorporate pre-treatment, post-treatment, and post-to-pre-treatment ratios of hematological parameters into response prediction.

Our study found that patients with an rCEA < 0.43 and rGGT ≥ 1.2 demonstrated better chemotherapy responses. The significance of CEA is well established, as approximately 70% of colorectal cancer patients exhibit elevated CEA levels. Stremitzer S et al. found that reductions in serum CEA levels could predict NAC efficacy in CRLM patients [21]. This is directly associated with the destruction of CEA-producing tumor cells by NAC, whether they are within lesions or circulating in the bloodstream. Previous studies have also shown that elevated postoperative CEA levels in CRLM patients undergoing curative hepatic resection are significantly associated with tumor recurrence [22]. This may result from the growth of microscopic lesions undetectable by imaging.

GGT is primarily located in the liver cell membrane and microsomes, participating in glutathione metabolism. GGT is abundant in the kidneys, liver, and pancreas, with serum GGT predominantly derived from the hepatobiliary system. Previous studies have indicated that elevated serum GGT levels are associated with poor prognosis in various cancers, including hepatocellular carcinoma, renal clear cell carcinoma, and gastric cancer. [23,24]. Additionally, cellular glutathione (GSH) is a primary defense against electrophilic agents such as cisplatin and alkylating agents [25]. GGT-mediated extracellular GSH metabolism provides cysteine for intracellular GSH synthesis, enabling GGT-high cells to utilize extracellular GSH more efficiently than control cells [26]. This indicates enhanced resistance to chemotherapy drugs, a phenomenon confirmed by multiple studies. In breast cancer, Sun et al. found that patients with high pre-treatment serum GGT levels exhibited lower sensitivity to NAC compared to those with low levels [27]. Similarly, in our CRLM cohort, higher pre-NAC GGT levels were associated with a lack of NAC response (Figure 4J).

Following the initiation of NAC, changes in serum GGT levels are influenced by multiple factors. GGT elevation is commonly observed in patients with hepatic steatosis and is considered a marker of liver damage [28]. Thus, changes in GGT may result from chemotherapy-induced liver function impairment. The use of irinotecan often leads to hepatocellular steatosis. However, as all our patients received the XELOX regimen, no elevation in other liver damage markers (e.g., ALT, AST, and TBIL) was observed after NAC (Figure 4K). Therefore, the post-treatment increase in GGT likely reflects the body’s response to chemotherapy and the tumor’s reaction to treatment. Previous studies have shown that as serum GGT levels increase, the incidence of grade 3–4 neutropenia post-NAC significantly decreases. This suggests that patients with higher serum GGT levels tolerate chemotherapy better than those with lower levels [27]. Additionally, studies have linked cisplatin-induced nephrotoxicity to oxidative stress [29]. In vitro studies demonstrated that GGT knockout or inhibition increased susceptibility to kidney injury following cisplatin treatment [30].

Although direct evidence is lacking, we hypothesize that patients who exhibit a favorable response to NAC with the XELOX regimen may experience an increase in GGT, thereby enhancing GSH utilization and gradually diminishing the efficacy of XELOX. When GGT synthesis in chemotherapy-sensitive tumor cells peaks, it may indicate the optimal cycle of NAC-induced tumor cell killing. Since our current study only utilized hematological data before and after NAC treatment, we could not validate this hypothesis. We plan to design a prospective validation cohort to collect GGT changes during NAC treatment, allowing for the dynamic prediction of NAC efficacy. Additionally, as targeted therapies are increasingly used in CRLM patients, we aim to evaluate the accuracy of our model across various NAC regimens.

The primary limitation of this study is the relatively small sample size. To ensure high-quality data, we rigorously selected patients with complete hematological data and reliable clinical and pathological information from extensive retrospective cohorts, minimizing the potential impact of this limitation on our findings. To the best of our knowledge, this is the first study to use pre- and post-NAC hematological data to predict treatment response in patients. The GCR score effectively predicted the treatment response of CRLM patients receiving the XELOX regimen. In the future, we aim to expand the sample size and evaluate the predictive performance of the GCR score in other treatment regimens.

## 5. Conclusions

The GCR index appears to be a promising biomarker for predicting the efficacy of NAC in CRLM patients. By comparing the changes in GGT and CEA levels before and after NAC, it can help forecast the tumor’s response to chemotherapy, thus preventing patients with poor NAC efficacy from undergoing premature surgery or receiving excessive NAC treatment. This provides a valuable tool for personalized therapy and prognostic assessment. Prospective studies are needed to further validate its application under different NAC regimens.

## Figures and Tables

**Figure 1 curroncol-32-00117-f001:**
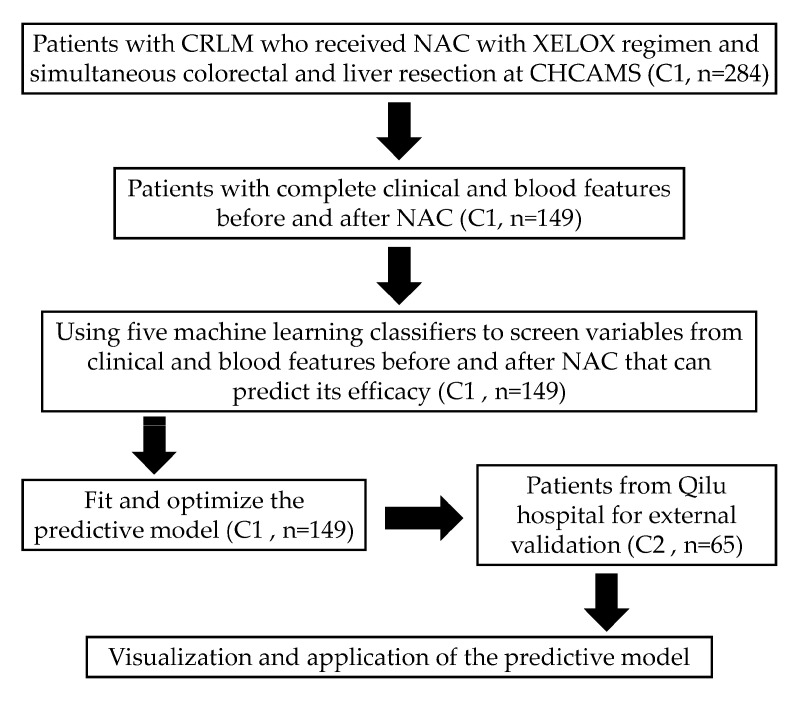
Flowchart of the study design. Abbreviations: CRLM: colorectal cancer liver metastases; CHCAMS: Cancer Hospital of Chinese Academy of Medical Sciences; NAC: neoadjuvant chemotherapy; C1 or C2: Center 1 or Center 2.

**Figure 2 curroncol-32-00117-f002:**
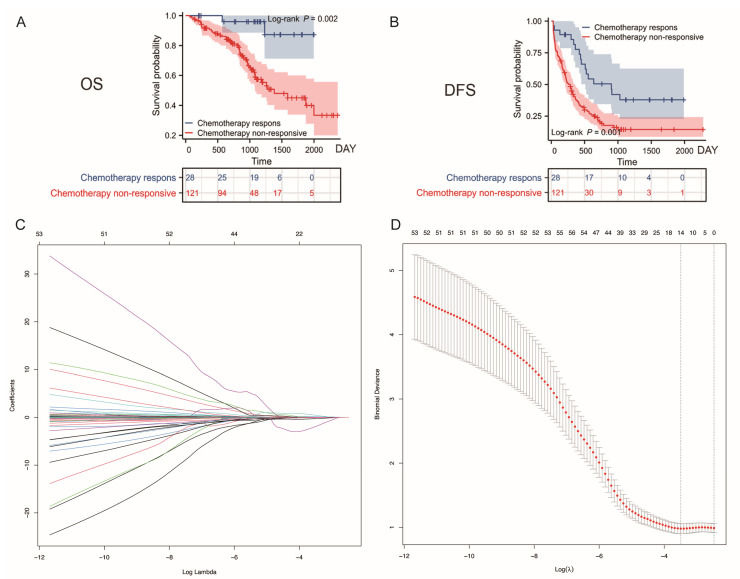
Lasso regression process for variable selection and the development of the binary logistic regression model. (**A**) Kaplan–Meier survival curves illustrating the relationship between chemotherapy response and overall survival of patients in Center 1. (**B**) Kaplan–Meier survival curves showing the relationship between chemotherapy response and disease-free survival in Center 1. (**C**) Tenfold cross-validation was employed to ensure the robustness of the model, with vertical lines drawn at selected feature values. (**D**) The coefficient profiles of 15 feature variables were derived from the log(λ) sequence in the LASSO model. Abbreviations: OS: overall survival; RFS: recurrence-free survival.

**Figure 3 curroncol-32-00117-f003:**
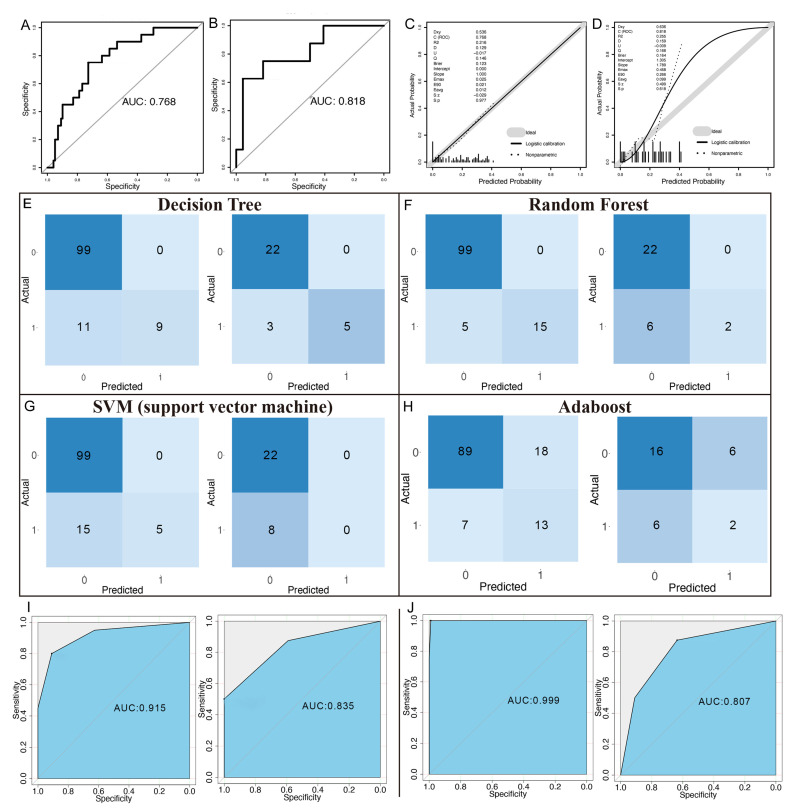
Survival curves and model evaluation. (**A**) AUC for the logistic model on the training dataset. (**B**) AUC for the logistic model on the validation dataset. (**C**) Calibration curve for the logistic model on the training set. (**D**) Calibration curve for the logistic model on the validation set. (**E**–**H**) Confusion matrices for the decision tree (**E**), random forest (**F**), support vector machine (**G**), and AdaBoost models (**H**). (**I**) ROC curves for the decision tree model for the training set (**left**) and validation set (**right**). (**J**) ROC curves for the random forest model for the training set (**left**) and validation set (**right**). Abbreviations: AUC: area under the curve.

**Figure 4 curroncol-32-00117-f004:**
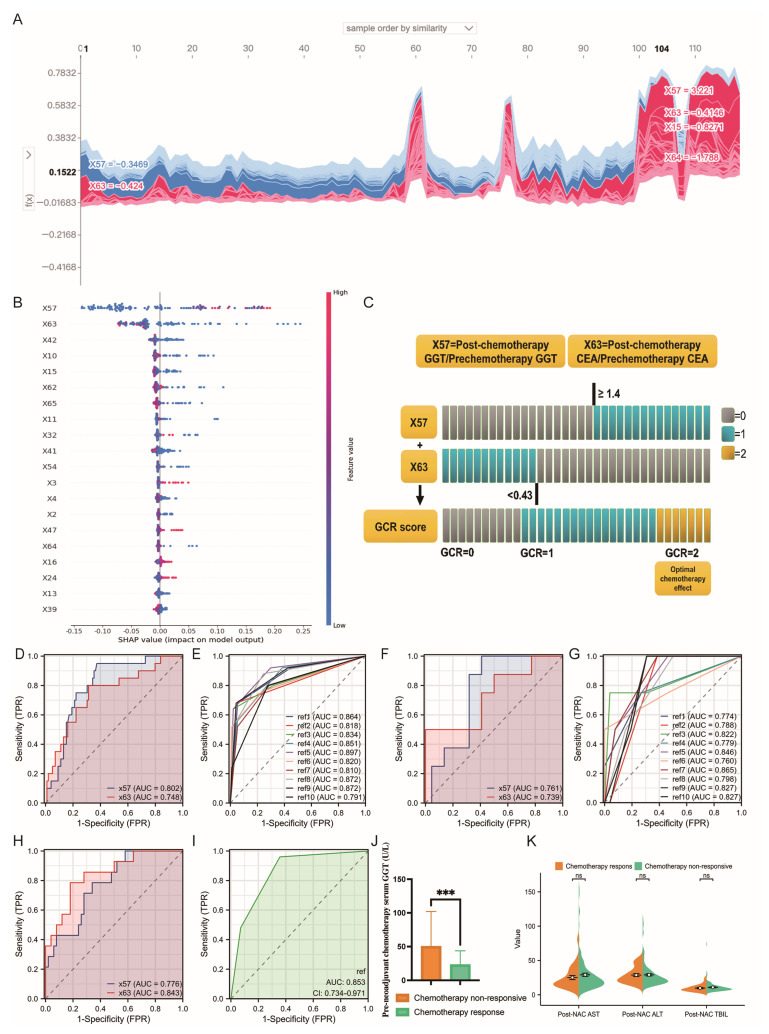
Interpretability and validation of models. (**A**): SHAP contribution of all features (ranked by feature similarity). The *X*-axis represents the number of samples, and the *Y*-axis shows the sum of SHAP values (calculated by multiplying each feature value by its SHAP value). Overall, X57 and X63 contribute the most to the model, with X57 having a negative impact and X63 having a positive impact. (**B**): SHAP-based feature importance in the random forest model. This figure illustrates the model’s point estimates. Each point represents a patient in the dataset, with colors indicating feature values: red for higher feature values and blue for lower ones. For instance, higher values of X57 (red) positively impact the model, while lower values of X63 (blue) also have a positive impact. (**C**): Visualization of the construction process of the best model (GCR model). (**D**): The ROC curves of variables X57 and X63 in the training set. (**E**): The 10-fold cross-validation ROC curve of the GCR predictive model in the training set. (**F**): The ROC curves of variables X57 and X63 in the validation set. (**G**): The 10-fold cross-validation ROC curve of the GCR predictive model in the validation set. (**H**): The ROC curves of variables X57 and X63 in the test set. (**I**): The ROC curve of the GCR predictive model in the test set. (**J**): The relationship between pre-NAC treatment GGT (bGGT) levels and chemotherapy response (*p* < 0.001). (**K**): The relationship between post-NAC treatment ALT (bALT), AST (bAST), and TBIL (bTBIL) levels and chemotherapy response (*p* > 0.05). *p* < 0.05; *p* < 0.01; *** *p* < 0.001. Abbreviations: ALT, alanine aminotransferase; AST, aspartate aminotransferase; TBIL, total bilirubin; GGT, gamma-glutamyl transferase; CEA, carcinoembryonic antigen; SHAP, Shapley additive explanations; FPR, false positive rate; TPR, true positive rate; X57, the ratio of GGT level after NAC to that before NAC (rGGT); X63, the ratio of CEA level after NAC to that before NAC (rCEA); ROC, receiver operating characteristic curve; AUROC, area under the ROC curve.

**Table 1 curroncol-32-00117-t001:** Comparison of clinical data among different centers.

Variables	Overall	Center 1	Center 2	*p*
	(n = 214)	(n = 149)	(n = 65)	
Age	57.5 (50.2, 64.0)	58.0 (52.0, 64.0)	55.0 (49.0, 63.0)	0.341
Gender				0.541
Female	71 (33.2%)	47 (31.5%)	24 (36.9%)	
Male	143 (66.8%)	102 (68.5%)	41 (63.1%)	
BMI	23.9 (22.0, 26.1)	23.9 (22.0, 26.1)	24.5 (22.3, 25.8)	0.410
CEA	7.6 (3.6, 23.7)	7.6 (3.5, 20.7)	8.1 (3.8, 37.8)	0.414 ^a^
Preoperative comorbidities				0.753
No	120 (56.1%)	82 (55.0%)	38 (58.5%)	
Yes	94 (43.9%)	67 (45.0%)	27 (41.5%)	
ASA Classification				0.188
I	9 (4.2%)	8 (5.4%)	1 (1.5%)	
II	180 (84.1%)	121 (81.2%)	59 (90.8%)	
III	25 (11.7%)	20 (13.4%)	5 (7.7%)	
CRS Score				0.051
1	21 (9.8%)	19 (12.8%)	2 (3.1%)	
2	62 (29.0%)	41 (27.5%)	21 (32.3%)	
3	113 (52.8%)	77 (51.7%)	36 (55.4%)	
4	16 (7.5%)	12 (8.1%)	4 (6.2%)	
5	2 (0.9%)	0 (0.0%)	2 (3.1%)	
Primary site				0.407
Colon	103 (48.1%)	75 (50.3%)	28 (43.1%)	
Rectum	111 (51.9%)	74 (49.7%)	37 (56.9%)	
Number of liver metastases				0.407
Solitary	111 (51.9%)	74 (49.7%)	37 (56.9%)	
Multiple	103 (48.1%)	75 (50.3%)	28 (43.1%)	
Differentiation				0.131
Well	144 (67.3%)	95 (63.8%)	49 (75.4%)	
Poor	70 (32.7%)	54 (36.2%)	16 (24.6%)	
T Stage				1.000
T1–T2	190 (88.8%)	132 (88.6%)	58 (89.2%)	
T3–T4	24 (11.2%)	17 (11.4%)	7 (10.8%)	
N Stage				0.396
N0	56 (26.2%)	42 (28.2%)	14 (21.5%)	
N1	158 (73.8%)	107 (71.8%)	51 (78.5%)	
Surgical sequence				0.342
Colorectal resection first	159 (74.3%)	114 (76.5%)	45 (69.2%)	
Liver resection first	55 (25.7%)	35 (23.5%)	20 (30.8%)	
Surgical procedures				0.180
Laparoscopy and laparoscopic assistance	32 (15.0%)	26 (17.4%)	6 (9.2%)	
Laparotomy	123 (57.5%)	90 (60.4%)	33 (50.8%)	
Laparoscopy	59 (27.6%)	33 (22.1%)	26 (40.0%)	
Blood loss (mL)	200.0 (200.0, 500.0)	200.0 (200.0, 400.0)	300.0 (200.0, 500.0)	0.400 ^a^
Operation time (min)	370.0 (300.0, 453.8)	370.0 (300.0, 450.0)	370.0 (300.0, 455.0)	0.761
Intraoperative transfusion				0.162
No	163 (76.2%)	118 (79.2%)	45 (69.2%)	
Yes	51 (23.8%)	31 (20.8%)	20 (30.8%)	
Postoperative complication				1.000
No	103 (48.1%)	72 (48.3%)	31 (47.7%)	
Yes	111 (51.9%)	77 (51.7%)	34 (52.3%)	
Postoperative adjuvant chemotherapy				0.856
No	82 (38.3%)	56 (37.6%)	26 (40.0%)	
Yes	132 (61.7%)	93 (62.4%)	39 (60.0%)	
Postoperative hospital stays (Day)	11.0 (9.0, 14.0)	10.0 (9.0, 14.0)	12.0 (10.0, 16.0)	0.827

^a^: Mann–Whitney U-test; BMI: body mass index; CEA: carcinoembryonic antigen; ASA: the American Society of Anesthesiologists; CRS: comprehensive risk score.

**Table 2 curroncol-32-00117-t002:** Testing efficiency of different machine learning models.

	Machine Learning Models	Accuracy	AUROC	Precision	Recall	F1-Score
Training Set	Decision Tree	0.908	0.915	1	0.45	0.621
	Random Forest	0.958	0.999	1	0.75	0.857
	Support Vector Machine	0.874	0.93	1	0.33	0.4
	Adaptive Boost	0.857	0.774	0.419	0.65	0.506
Validation Set	Decision Tree	0.9	0.835	1	0.625	0.769
	Random Forest	0.8	0.807	1	0.25	0.4
	Support Vector Machine	0.733	0.642	na	na	na
	Adaptive Boost	0.6	0.511	0.25	0.25	0.5

AUROC: area under the receiver operating characteristic curve; na: not available.

## Data Availability

The data presented in this study are available on request from the corresponding author.

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
