# Peer review of "Gamma-Glutamyl Transferase Plus Carcinoembryonic Antigen Ratio Index: A Promising Biomarker Associated with Treatment Response to Neoadjuvant Chemotherapy for Patients with Colorectal Cancer Liver Metastases"

_curroncol, 2025, doi:10.3390/curroncol32020117_

Round 1
Reviewer 1 Report
Comments and Suggestions for Authors
1. You mention "promising biomarker"—how did you determine it is promising?
2. You selected 21 blood features (BFs) from 76 characteristics, but in the results, you say 15 features were selected using Lasso. Why the difference? Why choose Lasso? Using at least 3 feature selection methods and comparing their classification accuracy would be better.
3. What do you mean by “Binary classification machine learning”? This term is not commonly used—please clarify.
4. In Figure 2, you show the coefficients and Parameters of the 15 selected features. It would be clearer to present them in a table instead of a figure.
5. Figure 3 has poor quality with small text and unnecessary items like E. Please improve its clarity.
6. The decision tree (DT) model is more interpretable, but why did you choose it over other models, especially since its performance is lower than random forest (RF)?
7. Your survival curves show significant differences, but it’s unclear how you controlled other patient characteristics besides chemotherapy response. Please make sure you address potential confounders.
8. You modified the TRG assessment tool to make it comparable between centers. How did you validate this modification to ensure consistency and avoid bias?
9. You mention no significant difference in baseline characteristics between the two centers (C1 and C2), but the patient populations might still differ. How did you control these differences?
10. You suggest that GGT (gamma-glutamyl transferase) might be involved in tumor defense mechanisms, but you haven’t validated this with real-time data or additional cohorts. Please clarify.
11. The study doesn’t discuss how targeted therapies might affect the outcomes. These therapies are becoming more common in CRLM treatment—how does this impact your findings?
12. The RECIST criteria do not account for tumor types, which could lead to discrepancies in response assessment. For example, tumors may shrink but remain highly active. Please explain how you addressed this limitation.
13. Your manuscript currently has a 36% similarity with other studies. Please revise the content to reduce this similarity to below 20%.
Reviewer 2 Report
Comments and Suggestions for Authors
The scientific work submitted to me for review addresses important issues of supporting the treatment of patients with colorectal cancer who have liver metastases. Below are some of my comments and suggestions for this work.
1) The authors of the manuscript use only the AUROC measure in the abstract. It would be good to also present the results for the classical classification accuracy. Such a result is easy for the reader to understand and directly speaks about the quality of the model.
2) In this article, I miss such a typical chapter related to the literature review. Since this article concerns patients with colorectal cancer, it would be good to refer to how machine learning already works in this subfield: in diagnosis, assessment of survival, selection of therapy. It is worth referring to the following articles:
https://www.mdpi.com/2072-6694/16/18/3205
https://www.nature.com/articles/s41598-024-77302-z
https://bmccancer.biomedcentral.com/articles/10.1186/s12885-023-10587-x https://www.nature.com/articles/s41698-024-00539-4
https://www.mdpi.com/2075-4418/13/2/301
3) The authors wrote:
We selected 21 blood features (BFs) from 76 blood characteristics including complete 98 blood counts, biochemical indicators, electrolytes, and coagulation function parameters.
Then the authors write:
In the training and validation sets, we performed 10-fold cross-validation using LASSO regression analysis on 63 blood features and 7 clinical characteristics available be fore chemotherapy. The minimum mean square error was 0.94 (Figure 2C, D), resulting in the selection of 15 important feature variables (Figure 2E)
How were these features selected? Was it a matter of expert knowledge or were some algorithmic feature selection methods used? What features were ultimately selected?
4) Among the selected classification models, I miss the use of XgBoost or LightGBM algorithm. These algorithms generally have the highest classification accuracy on tabular data.
5) It might be worth splitting Figure 2 into several smaller figures. This will increase readability.
6) Similarly, Figure 3 is not very legible. It may be worth limiting ourselves to the error matrix of the best model only.
7) How did the unbalanced data set affect the results achieved?
8) Figure 4 is also difficult to analyze.
9) The "Conclusions" section is very limited. Please expand it.
